# Development and Application of an Optogenetic Manipulation System to Suppress Actomyosin Activity in *Ciona* Epidermis

**DOI:** 10.3390/ijms24065707

**Published:** 2023-03-16

**Authors:** Jinghan Qiao, Hongzhe Peng, Bo Dong

**Affiliations:** 1Fang Zongxi Center, MoE Key Laboratory of Marine Genetics and Breeding, College of Marine Life Sciences, Ocean University of China, No. 5 Yushan Road, Qingdao 266003, China; 2Laoshan Laboratory, Qingdao 266237, China; 3Institute of Evolution & Marine Biodiversity, Ocean University of China, Qingdao 266003, China

**Keywords:** *Ciona*, optogenetics, actomyosin contractility, epidermal cells

## Abstract

Studying the generation of biomechanical force and how this force drives cell and tissue morphogenesis is challenging for understanding the mechanical mechanisms underlying embryogenesis. Actomyosin has been demonstrated to be the main source of intracellular force generation that drives membrane and cell contractility, thus playing a vital role in multi-organ formation in ascidian *Ciona* embryogenesis. However, manipulation of actomyosin at the subcellular level is impossible in *Ciona* because of the lack of technical tools and approaches. In this study, we designed and developed a myosin light chain phosphatase fused with a light-oxygen-voltage flavoprotein from *Botrytis cinerea* (MLCP-BcLOV4) as an optogenetics tool to control actomyosin contractility activity in the *Ciona* larva epidermis. We first validated the light-dependent membrane localization and regulatory efficiency on mechanical forces of the MLCP-BcLOV4 system as well as the optimum light intensity that activated the system in HeLa cells. Then, we applied the optimized MLCP-BcLOV4 system in *Ciona* larval epidermal cells to realize the regulation of membrane elongation at the subcellular level. Moreover, we successfully applied this system on the process of apical contraction during atrial siphon invagination in *Ciona* larvae. Our results showed that the activity of phosphorylated myosin on the apical surface of atrial siphon primordium cells was suppressed and apical contractility was disrupted, resulting in the failure of the invagination process. Thus, we established an effective technique and system that provide a powerful approach in the study of the biomechanical mechanisms driving morphogenesis in marine organisms.

## 1. Introduction

Morphogenesis, as a basis of embryo development, is a pattern formation process via cell and tissue reshaping and migration. The morphogenetic movement can be regulated by the genetic blueprint and molecular signaling network, but ultimately, biomechanical force is required [1,2,3,4]. How biomechanical forces are precisely generated to drive cell reshaping is a hot topic in developmental and cell biology. Many studies have demonstrated that actomyosin is an important source of intracellular mechanical force that drives cell contractility [5,6,7,8]. Actomyosin, which is enriched in the cortex of the membrane, is closely related to cell contractility and cell morphology. Non-muscle myosin II (NM II), consisting of two heavy chains, two light chains, and two regulatory light chains (RLC) [9,10], is the main myosin component involved in non-muscle cell contraction [9,11]. The phosphorylation of myosin light chain (MLC) by myosin light chain kinase (MLCK) and Rho-kinase (ROCK) is essential for myosin function [12,13]. Myosin light chain phosphatase (MLCP) can dephosphorylate MLC [14,15]. On the other hand, RhoA-ROCK signaling can abolish the activity of MLCP by inhibiting the phosphorylation of its myosin phosphatase target subunit (MYPT) [10,12]. However, how actomyosin localizes at the specific subcellular position and accurately regulates cell contractility remain unclear.

In marine organisms, the diversity of species and morphogenesis processes provides abundant experimental materials for biomechanical force research [16]. For example, during the development of the *Ciona* embryonic notochord, actomyosin asymmetrically localizes on the ventral side of the notochord, providing the force for tail bending [17,18]. In addition, during the endodermal invagination process in *Ciona*, actin sequentially localizes on the apical and basolateral cell interfaces to drive apical contraction and invagination [19,20]. At present, some low-molecular-weight compounds are widely used to inhibit the activity of NM II, such as blebbistatin [21], ROCK inhibitor Y-27632 [22], MLCK inhibitor ML-7 [23], etc. The function of actomyosin can also be blocked by myosin light chain dominant negative mutation [24]. However, disadvantages of these methods limit their applied range. The inhibition of myosin via inhibitor treatment or dominant negative mutation shows non-specificity at the subcellular level, it is difficult to eliminate in a short time period, and there can be side effects or toxicity on tissues and cells. Therefore, a faster and controllable approach to regulate actomyosin activity at the subcellular level is urgently required. The development of optogenetics provides us with new opportunities for technological innovation in the *Ciona* model system.

Optogenetics is a technology that combines optics and genetics to control the function of proteins with light. Due to the rapid switching and ease of subcellular manipulation, optogenetics can control proteins and subcellular localization with unprecedented spatial and temporal resolution [25]. The concept can be traced back to the 1970s when Francis Crick noted that an important challenge facing neuroscience was how to precisely control the activity of a single type of cell without affecting other cells, suggesting that light could be a key tool in this process but the method was not found at that time [26]. At the same time, bacteriorhodopsin, a light-sensitive ion pump, was identified [27,28]. With the development of science, scientists have identified some similar proteins that receive light to transport various ions [29,30]. However, it was not until 2005 that anyone achieved millisecond control of neurons using members of the rhodopsin family [26,31,32,33]. The spatiotemporal regulation of signaling molecules is one of the focuses of developmental biology research and controlling membrane potential alone is not sufficiency to meet research requirements [34]. To this end, scientists have developed a series of photosensitive proteins that have different chromophores and exhibit different properties under the stimulation of different wavelengths of light, such as light-oxygen-voltage (LOV), UV RESISTANCE LOCUS 8 (UVR8), blue light–using flavin (BLUF), Opsin [34,35,36,37], etc. Based on the properties of different photosensitive proteins, researchers can design a variety of systems in developmental biology to control the localization, clustering, sequestration, and release of regulatory signaling molecules.

Among these known photosensitive proteins, the LOV system has been engineered into various structures and widely used [38,39]. The LOV photoreceptors of these systems consist of sensor and effector domains, and these two parts are usually connected by an α-helix domain [40]. Under blue light, the LOV photoreceptors detect and absorb optical energy and then undergo structural changes that alter their biological function [41,42]. Specifically, the LOV domain belongs to the Per ARNT-Sim (PAS) domain superfamily, which is about 110 amino acids long and carries a flavin chromophore. These cofactors noncovalently bind to the surrounding LOV domain in the dark and form covalent bonds with a cysteine residue after absorbing blue light, thus causing the α-helix to change conformation and unfold. Moreover, when the blue light was removed, they spontaneously decayed back into the state of darkness [41,43]. Based on LOV, scientists have built a series of optogenetic systems that control tissue patterns through regulation of signaling molecules, gene expression, and cell motility [34,44,45,46]. One of them, the photosensitive protein BcLOV4 from *Botrytis cinerea*, can rapidly bind to the plasma membrane through electrostatic interaction under blue light [38,44,46,47,48]. In previous studies, a method has been established to change the phosphorylation level of myosin in cells by altering the localization of MLCP using optogenetics [45]. However, the localization of iLID-GFP proteins could not be examined before illumination because of the similar wavelength (488 nm) between GFP excitation and this system. In addition, we found that this system was highly light-sensitive and easily illuminated under the confocal microscopy when we tested it in *Ciona* system.

In this study, we modified MLCP and established an optogenetic system using the photosensitive protein BcLOV4, which was named the MLCP-BCLOV4 system. We validated this system in HeLa cells and *Ciona* embryos and larvae. Finally, we applied this system to interfere with the localization of phosphorylated myosin in apical contraction during the morphogenesis of the *Ciona* atrial siphon. We eventually applied this technique to control the activity of epidermal myosin and cell contractility of *Ciona*, which provides a powerful tool for the study of the biomechanical forces driving morphogenesis in marine organisms.

## 2. Results

### 2.1. Design of an MLCP-BcLOV4 System to Suppress the Activity of Myosin II at the Cellular Level

MLCP, as a phosphorylase kinase regulated by the RhoA/ROCK signaling pathway (Figure 1a), consists of a regulatory subunit MYPT1, a catalytic subunit PP1c, and a small subunit M20 [13,49]. MYPT includes the PP1C binding domain, phosphorylated MLC binding domain, myosin binding domain, and M20 binding domain (Figure 1b). It has been reported that the 1–38 amino acids (a.a.) at the N-terminus of MYPT are required for the functioning of PP1c, and the 170–296 a.a. of MYPT comprise the phosphorylated MLC binding domain [12,49]. In addition, the 1–169 a.a. of MYPT have high light control efficiency as a regulatory element, which has been used in conjunction with other optogenetic systems (iLID/SspB system) to reduce contractile force in mammalian cells and *Xenopus* embryos [45]. Based on this knowledge, we designed the combination of PP1c and MYPT (1–169 a.a.), which serves the function of myosin dephosphorylation but lacks the spontaneous myosin-binding activity. The PP1-MYPT (1–169 a.a.) could act as a myosin inhibiting factor only when it is artificially docked to phosphorylated myosin. Then, we linked the PP1c-MYPT (1–169 a.a.) to the photosensitive protein BcLOV4 and constructed the MLCP-BcLOV4 system. Because of the plasma membrane-binding activity of BcLOV4 via electrostatic protein-lipid interaction under 488 nm blue light [46], the PP1c-MYPT (1–169 a.a.) could spatially localize sufficiently close to the myosin II in the cell cortex, resulting in endogenous dephosphorylation of MLC and release of the contractility at the cell cortex (Figure 1c). To visualize the localization of PP1c-MYPT (1–169 a.a)-BcLOV4, mCherry was linked at the C-terminus of the construct. However, similar to previous studies [45,50], we found that the fusion protein was localized at the cell nucleus regardless of light exposure (Figure 1d). To solve this problem, a nuclear export signal (NES) was added into the C-terminus of the fusion protein. We then found that NES worked and the construct was located in the cytoplasm after light exposure (Figure 1d). We further found high sequence identities of human MYPT (1–169 a.a) and PP1C with those in the *Ciona* genome (Figure 1e), which provided support that the MLCP-BcLOV4 system could be tested in *Ciona* living cells.

### 2.2. Validation of the MLCP-BcLOV4 System in HeLa Cells

We first tested the excitation effect of the MLCP-BcLOV4 system with the CMV promoter in HeLa cells with gradient laser intensities. The results showed that 0.5% 488 nm laser confocal microscope (ZEISS LSM 980) was the minimum intensity to activate the MLCP-BcLOV4 system’s membrane location (Figure 2a), indicating the high sensitivity of the MLCP-BcLOV4 system to blue light.

Then, we tested the working effects of the MLCP-BcLOV4 system in Hela cells. The MLCP-BcLOV4 construct (CMV>PP1c::MYPT169::BcLOV4::mCherry::NES) and the control (CMV>PP1c::MYPT169::mCherry::NES) were transfected into HeLa cells, respectively, and cultured in a dark incubator. Then, we measured the fluorescence intensity of the cell membrane and cytoplasm before and after laser irradiation with 488 nm blue light. Under dark conditions, mCherry signals dispersed in the cytoplasm in both the MLCP-BcLOV4-expressed and MLCP-expressed groups. After 15 min of blue light irradiation, the mCherry signal intensity significantly increased at the cell membrane in the MLCP-BcLOV4-expressed group, while it decreased in the cytoplasm. No significant change was observed in the MLCP-expressed group (Figure 2b,c). In addition, we quantified the membrane-to-cytoplasm ratio of the mCherry signal intensity in the MLCP-BcLOV4 system before and after 0.5% blue light exposure for 15 min. The statistic results indicated that the membrane-to-cytoplasm ratio of the mCherry signal intensity significantly increased in the experimental group compared with the control group (Figure 2d). These results suggested that the MLCP-BcLOV4 system had high light-dependent membrane localization efficiency in HeLa cells.

Next, we examined the effects of the MLCP-BcLOV4 system on the suppression of myosin activity and inhibition of cell contractility in HeLa cells. We measured and quantified the circularity (ratio of the perimeter to the square root of its area) of the cell section through the nuclear geometric center, reflecting myosin activity and cell contract. To reduce the effect of cell connections on cell morphology, individual cells without contact with other cells were chosen for the measurement (Figure 2e and Figure A1). The results showed that after 30 min of blue light exposure, the circularity of cells in the MLCP-BcLOV4-expressed group significantly decreased compared with that in the MLCP-expressed group. This result indicated that the MLCP-BcLOV4 system could effectively suppress cell contractility, which lead to cell reshaping in HeLa cells.

### 2.3. MLCP-BcLOV4 System Suppressed the Contractility of Epidermal Cells in Ciona Embryos

To test the efficiency of the MLCP-BcLOV4 system in vivo, we next expressed MLCP-BcLOV4 and MLCP-only driven by an epidermis-specific promoter *Epi1* in *Ciona* embryos. An F-actin marker lifeact-eGFP was co-expressed in the *Ciona* epidermis to visualize the cell boundary [51,52,53,54,55]. We measured the fluorescence intensity of the cell membrane and cytoplasm before and after irradiating with 488 nm blue light. The results showed that the mCherry signals were dispersed in the cytoplasm of both the MLCP-BcLOV4-expressed group and the MLCP-expressed group under dark conditions. After 15 min of blue light irradiation, the signal intensity of mCherry on the cell membrane in the MLCP-BcLOV4-expressed group significantly increased, while there was no significant change in the MLCP-expressed group (Figure 3a,b). Next, we compared the membrane-to-cytoplasm ratio of the mCherry signal intensity in the MLCP-BcLOV4-expressed group and MLCP-expressed group before and after 15 min of blue light irradiation. It can be seen that the membrane-to-cytoplasm ratio of the mCherry signal intensity in the MLCP-BcLOV4-expressed group was higher than that before light irradiation, while there was no significant change in the MLCP-expressed group (Figure 3c). This result indicated that the MLCP-BcLOV4 system had a better blue light-dependent membrane localization effect in the epidermis cells of *Ciona* embryos.

Because of the cell junction and surrounding basal membrane, the mechanical action of epidermal cells is more complex than that of cultured cells. Therefore, we measured and quantified the length changes of the cell membrane (boundary change ratio) of *Ciona* epidermal cells in the MLCP-BcLOV4-expressed and MLCP-expressed groups, respectively. The results exhibited that the cell boundary length of the MLCP-BcLOV4-expressed group significantly increased after 30 min of blue light exposure compared to the MLCP-expressed group (Figure 3d–f), indicating a defect in cytoskeletal contractility in the MLCP-BcLOV4-expressed group. This result suggested that the MLCP-BcLOV4 system could effectively suppress the cell contractile force in the epidermis of *Ciona* embryos.

### 2.4. MLCP-BcLOV4 System Disrupted Atrial Siphon Invagination via Abolishing the Apical Contraction in Ciona Larvae

The formation of the atrial siphon in *Ciona* experiences an apical contraction in the central epidermal cell and subsequent invagination. The immunofluorescence experiment exhibited the enrichment of active myosin II in the apical surface of atrial siphon primordium cells compared to that in the basal surface (Figure 4a), which gave us a chance to test whether the MLCP-BcLOV4 system can effectively change the localization and activity of myosin in organ morphogenesis during *Ciona* larval development.

We expressed the MLCP-BcLOV4 system in *Ciona* larval siphon cells with the *Epi1* promoter. After 488 nm blue light exposure to activate the system, the larvae were fixed and the active myosin in atrial siphon primordium cells was examined using the active myosin II (S19) antibody (Figure 4b). After imaging, we intercepted the side section through the center of the atrial siphon invagination (Figure 4d). Then, we measured the myosin II signal on the apical and basal cell boundaries of both MLCP-BcLOV4 and wild-type siphon cells. The results showed that the activity of phosphorylated myosin was suppressed in the MLCP-BcLOV4-expressed siphon cells (Figure 4c and Figure A2). Our results demonstrated that the MLCP-BcLOV4 system we developed could control the activity of epidermal myosin and cell contractility in *Ciona*.

## 3. Discussion

In this study, we designed and developed a MLCP-BcLOV4 system to regulate the activity and polarity of myosin II and the cell contractility of the *Ciona* epidermis. This system was then successfully applied to interfere with the localization and polarity of phosphorylated myosin in apical contraction during the morphogenesis of the *Ciona* siphon. Our results demonstrated that the MLCP-BcLOV4 system is a powerful tool for the study of biomechanical mechanisms during organ morphogenesis compared with chemical inhibitor [21,22,23] and dominant negative mutation approaches [24,56], which have high inhibition efficiency on the activity of myosin but still have limitations on the working range and time. The MLCP-BcLOV4 system could rapidly change the localization and polarity of myosin II at the subcellular level, which greatly improved the accuracy and reduced the potential side effects of the experimental manipulation.

Several optogenetic systems have been established to regulate myosin activity at the cell and tissue levels. For example, optogenetic systems were applied to control RhoA activity [36,44,46]. Light-dependent recruitment of RhoGEF triggers RhoA activation, which induces myosin phosphorylation and increases the cell contractile force [57]. In addition, the OptoMYPT system has been utilized in mammalian cells and *Xenopus* embryos to reduce the cell contractile force [45]. However, so far, none of optogenetic manipulation systems were established and applied to ascidian and marine animals. Marine organisms provide a series of biomaterials for the study of developmental biology and biomechanics because of the diversity of their morphogenesis [16,58]. For example, during the morphogenesis of the *Ciona* embryonic notochord, the asymmetrically localization of actomyosin on the ventral side of notochord provides the force for tail bending [17,18,59]. During the endodermal invagination process of *Ciona*, actin sequentially localizes on the apical surface to drive apical contraction and invagination [19,20]. In addition, the morphogenesis of the *Ciona* atrial siphon is a typical example of invagination caused by apical contraction (Figure 4). The MLCP-BcLOV4 system is the first one that can be applied to marine organisms. It exhibited a good ability to regulate myosin activity in *Ciona* epidermis cells, providing a powerful tool to help answer developmental and cell biology questions, especially those about the biomechanical mechanisms of organ morphogenesis.

There are several advantages of this system. Firstly, the MLCP-BcLOV4 system has subcellular level resolution, which enabled us to achieve precise control of the localization of protein and reduce the cell contractile force. This system allowed us to make not only cell-to-cell comparisons but also intracellular comparisons to explore how regional aggregation of myosin affects cellular behavior, which is particularly important in the study of some biological processes involved in the polarized distribution of phosphorylated myosin, such as the apical contraction process during *Ciona* siphon morphogenesis. Secondly, the MLCP-BcLOV4 system can be quickly and repeatedly activated. Taking advantage of this property, we could repeatedly relax the cell contractile force. Instantaneous stimulation can observe how cells respond to changes in mechanical force without impeding normal tissue development. This transient perturbation can be used to explore the direct effects of perturbation rather than the indirect effects [34]. Finally, using a tissue-specific driver, the system realized tissue-specific manipulation to minimize the impact on other tissues during experiments in stereoscopic embryos. Therefore, we could explore the mechanical interactions between multiple tissues.

Together, we designed and established an optogenetic manipulation MLCP-BcLOV4 system to regulate the activity and polarized distribution of myosin II. We have demonstrated that the system could interfere with the polarity of phosphorylated myosin II in apical contraction during *Ciona* atrial siphon morphogenesis, leading to failure of epidermal cell invagination. Our technique is expected to be a powerful tool for studying the unique morphogenesis of marine organisms.

## 4. Materials and Methods

### 4.1. Experimental Animal Preparation and Electroporation

*Ciona* adults were collected from the sea area of Qingdao and Rongcheng, Shandong, China. They were maintained in the laboratory seawater circulation system under constant light to stabilize their state. Mature eggs and sperm were collected from adults separately after dissection and then mixed in vitro for fertilization. Fertilized eggs were dechorionated in seawater containing 1% sodium thioglycolate (T0632; Sigma), 0.05% protease E (P5147; Sigma), and 0.032 M NaOH. Dechorionated eggs were used for plasmid electroporation [60]. Finally, they were cultured in an agar-coated dish with microporous-filtered seawater (MFSW) at 18 °C for further observation.

### 4.2. Plasmids Construction

The BcLOV4-mCherry module was amplified with primers 5′-ATGGCCACAGACGCAATCG-3′ and 5′-GATTATGATCTAGAGTCGCGGCCGC-3′ from opto-RhoA-mCherry_pcDNA3.1 [46] (Addgene plasmid # 164472). The PP1c module was amplified with primers 5′-ATGGCGGACGGGGAGC-3′ and 5′-CCTTTTCTTCGGCGGA-3′. The MYPT169 module was amplified from HeLa cDNA with primers 5′-ATGAAGATGGCGGACGCG-3′ and 5′-AGCTGCTTCTATATCAACCCCTTG-3′. These modules were subcloned into pEGFP-C1 to construct the optogenetic expression vector.

### 4.3. Cell Transfection

HeLa cells in this program were purchased form the National Collection of Authenticated Cell Culture (https://www.cellbank.org.cn, accessed on 22 June 2021). Cell transfection experiments were performed using Lipofectamine™ 3000 Transfection Reagent (L3000015; Thermo Fisher). After transfection, the cells were cultured overnight in absolutely dark conditions until observation. Then, the cells were placed in a 5% CO_2_ humidified incubator at 37 °C in the dark for 12 h.

### 4.4. Immunofluorescence

*Ciona* embryos were fixed with stationary liquid [19] (100 mM HEPES, pH = 6.9; 100 mM EGTA, pH = 7.0; 10 mM MgSO_4_; 2% formaldehyde; 0.1% glutaraldehyde; 300 mM dextrose; and 0.2% Triton X-100) for 40 min at room temperature. After three washes with PBS, the embryos were incubated in PBST (PBS with 0.1% Triton X-100) for 30 min to achieve permeabilization. Then, the embryos were treated with 0.1% sodium borohydride in PBS for 20 min at room temperature. Next, a 1:250 dilution of phospho-myosin light chain 2 (Ser19) antibody (#3671; Cell Signaling) was added and incubated at room temperature for 24 h. After three washes with PBS, a 1:200 dilution of Alexa Fluor 568 anti-Rabbit IgG (A11011; Invitrogen) was added and incubated for 48 h at room temperature. As needed for cell boundary visualizing, a 1:200 dilution of Alexa Fluor 488 Phalloidin (A12379; Invitrogen) was added. Finally, after three washes, the embryos were coated with mounting medium with DAPI and mounted for further observation.

### 4.5. Imaging and Optogenetics

Live imaging, photo-activation experiments, and image acquisition were carried out using a Zeiss LSM 980 (Carl Zeiss). The nominal power of the diode laser 488 nm was 30 mW. Living HeLa cells and *Ciona* embryos were placed in a 35 mm glass-bottom dish for imaging. To prevent unnecessary photoactivation, a plasmid with mCherry was co-electroporated to locate the embryos and select the test sites. Then, the selected testing area was exposed to the 488 nm laser to activate the optogenetic system.

## Figures and Tables

**Figure 1 ijms-24-05707-f001:**
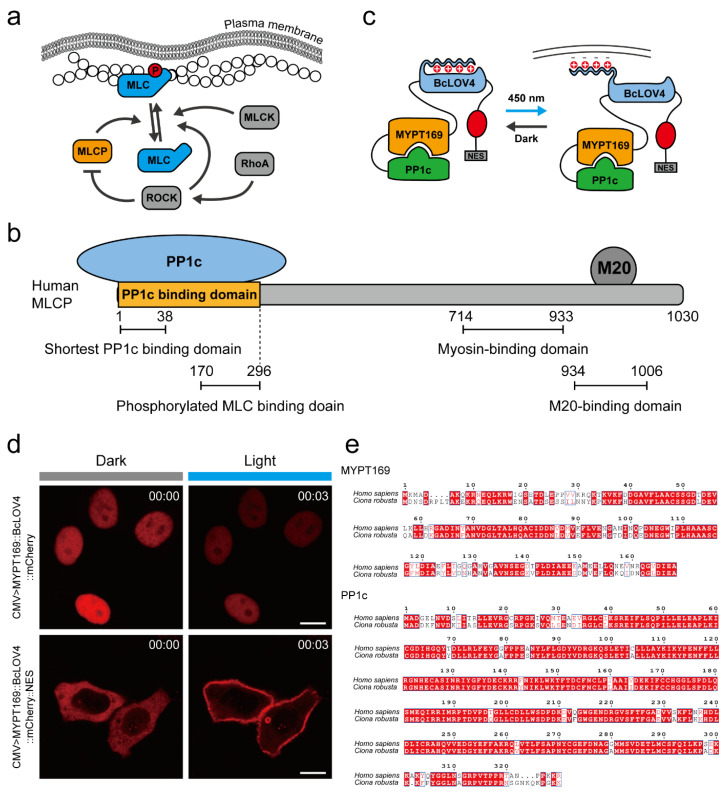
Design of an MLCP-BcLOV4 system to suppress the activity of myosin II at the cellular level. (**a**) Schematic illustration of myosin phosphorylation regulatory network; (**b**) Domain structure of MLCP holoenzyme; (**c**) Schematic illustration of the structure and working principle of the MLCP-BcLOV4 system. PP1C-MYPT169-BcLOV4-mCherry-NES disperses in the cytoplasm. Upon blue light illumination, BcLOV4 binds to the plasma membrane through electrostatic interaction, and the fused MYPT169 and PP1C are transferred to the plasma membrane to play their corresponding roles; (**d**) Representative images of localization of MYPT169-BcLOV4-mCherry and MYPT169-BcLOV4-mCherry-NES in HeLa cells in darkness and after 3 min of illumination under 488 nm blue light. Scale bar, 10 μm; (**e**) Sequence alignment of MYPT169 and PP1C in human and *Ciona* genomes.

**Figure 2 ijms-24-05707-f002:**
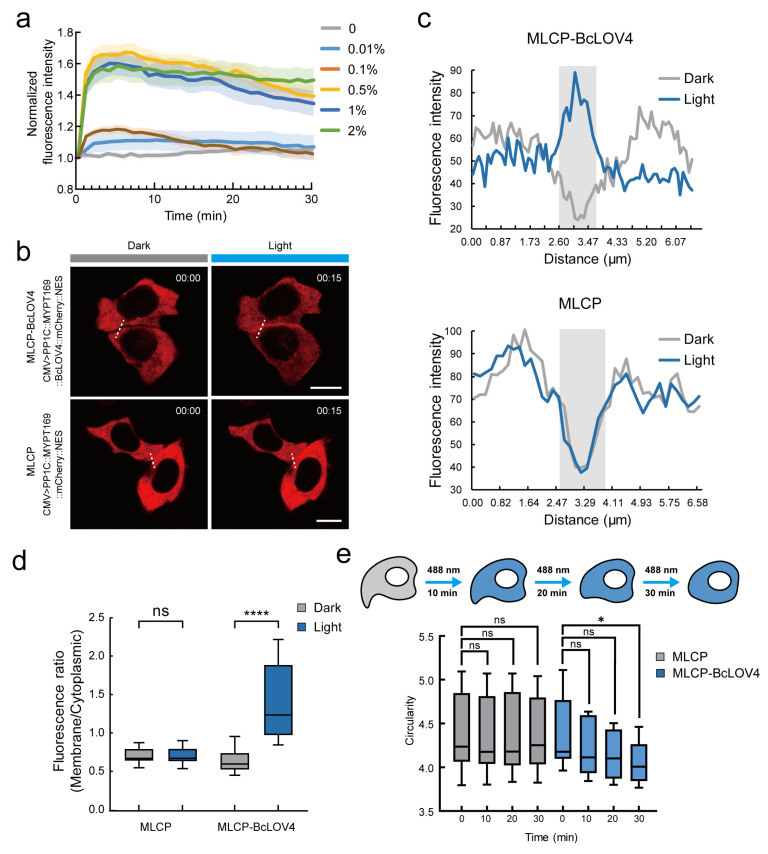
Validation of the MLCP-BcLOV4 system in HeLa cells. (**a**) Quantitative statistics of fluorescence intensity change of HeLa cell membrane in the MLCP-BcLOV4-expressed group within 30 min under stimulation of blue light at 0 to 2% intensity. The solid line and pale area are normalized statistics of the mean value and SEM of the change in the fluorescence intensity of the cell membrane. *n* = 10 cells; (**b**) Representative images of the expression of the MLCP-BcLOV4-expressed group (CMV>PP1c::MYPT169::BcLOV4::mCherry::NES, top row) and the MLCP-expressed group (CMV>PP1c::MYPT169::mCherry::NES, bottom row) in HeLa cells. The left is the image in the dark and the right is the image after 15 min of illumination under 488 nm blue light. Scale bar, 10 μm; (**c**) Quantitative statistics of fluorescence intensity at the white dotted line in panel b. The gray line is the statistical graph in the dark state and the blue line is the statistical graph after 15 min of blue light irradiation. The light gray area is the cell boundary; (**d**) Quantification of membrane-to-cytoplasm ratio of mCherry signal fluorescence ratios of the MLCP-BcLOV4-expressed group and the MLCP-expressed group after 15 min of exposure to 0.5% blue light. ****, *p* < 0.0001 (student’s *t*-test). *n* = 20 cells; (**e**) Schematic diagram of blue light treatment of HeLa cells with MLCP-BcLOV4 system. The circularity (ratio of the perimeter to the square root of its area) of the cell section through the nuclear geometric center in the MLCP-BcLOV4-expressed group and MLCP-expressed group changed at 10, 20, and 30 min after blue light stimulation. *, *p* < 0.05 (student’s *t*-test). *n* = 10 cells.

**Figure 3 ijms-24-05707-f003:**
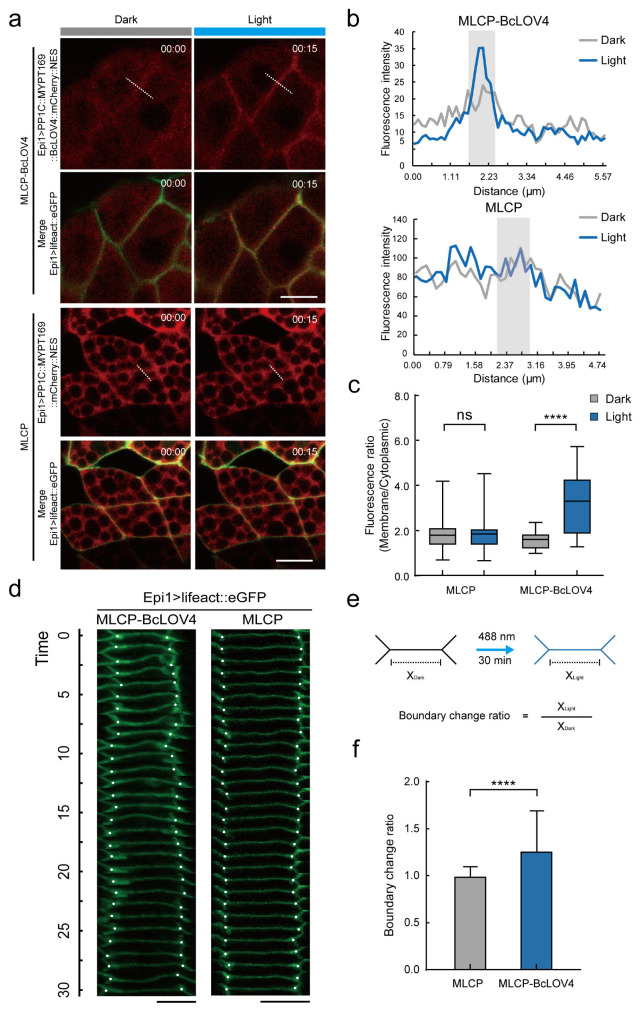
MLCP-BcLOV4 system suppressed the contractility of epidermal cells in *Ciona* embryos. (**a**) Representative images of the expression of the MLCP-BcLOV4-expressed group (Epi1>PP1c::MYPT169::BcLOV4::mCherry::NES, top two rows) and the MLCP-expressed group (Epi1>PP1c::MYPT169::mCherry::NES, bottom two rows) in the epidermis of *Ciona* embryos. On the left are images in the dark and on the right are images after 15 min of illumination under 488 nm blue light. Scale bar, 5 μm; (**b**) Quantitative statistics of fluorescence intensity at the white dotted line in panel a. The gray line is the statistical graph in the dark state and the blue line is the statistical graph after 15 min of blue light irradiation. The light gray area is the cell boundary; (**c**) Quantification of membrane-to-cytoplasm ratio of mCherry signal fluorescence of the MLCP-BcLOV4-expressed group and the MLCP-expressed group after 15 min of exposure to 0.5% blue light. ****, *p* < 0.0001 (student’s *t*-test). *n* = 20 cells; (**d**) Representative images of epidermal cell boundary length over time in *Ciona* embryos from the MLCP-BcLOV4-expressed group and MLCP-expressed group during a 30 min period. White dots show the location of tricellular junctions. Scale bar, 5 μm; (**e**) Schematic diagram of boundary change ratio calculation; (**f**) The boundary change ratio of epidermal cells in the MLCP-BcLOV4-expressed group and MLCP-expressed group after 30 min of blue light irradiation. ****, *p* < 0.0001 (student’s *t*-test). *n* = 100 cells.

**Figure 4 ijms-24-05707-f004:**
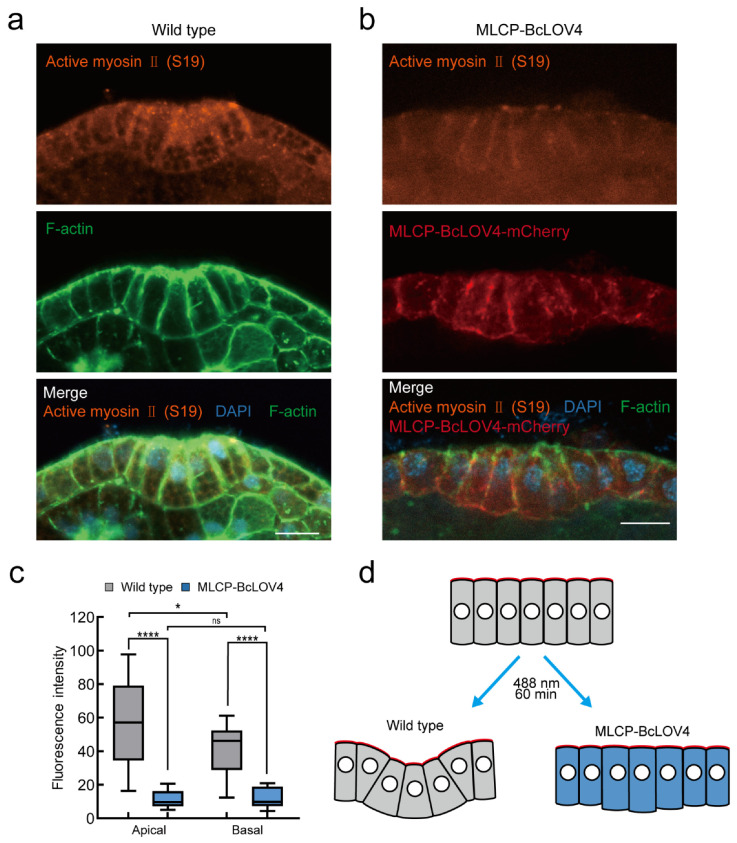
MLCP-BcLOV4 system disrupted atrial siphon invagination via abolishing the apical contraction in *Ciona* larvae. (**a**) Representative image of myosin immunofluorescence assay of *Ciona* atrial siphon in wild type. Scale bar, 10 μm; (**b**) Representative image of myosin immunofluorescence assay of *Ciona* atrial siphon in MLCP-BcLOV4 system group. Scale bar, 10 μm; (**c**) Quantification of fluorescence intensity of active myosin II signal of wild type and MLCP-BcLOV4-expressed group after 60 min of exposure to 0.5% blue light. *, *p* < 0.05, ****, *p* < 0.0001 (student’s *t*-test). *n* = 14 cells; (**d**) Schematic diagram of blue light treatment of *Ciona* atrial siphon of wild type and MLCP-BcLOV4-expressed group. Red indicates the apical surface of the cell.

## Data Availability

All data generated or analyzed during this study are included in the manuscript and Appendix A.

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
