# Peer review of "Development and Application of an Optogenetic Manipulation System to Suppress Actomyosin Activity in Ciona Epidermis"

_ijms, 2023, doi:10.3390/ijms24065707_

Round 1

Reviewer 1 Report

Optogenetic method utilized to manipulate subcellular protein activities is of great interest. The authors of this manuscript developed a new optogenetic system to manipulate the MLCP activity and thus suppress actomyosin activity in Ciona epidermis. The overall story is very interesting except there are several points which need to be improved.

First, as already mentioned by the authors, there is already a similar system using iLID to manipulate MLCP activity. A question to the audience might be, why the authors do not test the already established optogenetic system based on iLID in Ref 45 but develop a new system with BcLOV4 in this study. Obviously, there are several differences between iLID and BcLOV4, the differences should be introduced in more detail, this will help the audience to choose the potential tool for their own research.

Second, Figure 2e is somehow not very convincing as it shows in its current form. It is not very common to label p<0.1 as “ * “, which usually means significant. For such a result, it is also not very convincing to me with n=10. Since the cells should vary a lot even in their natural condition, how choosing the 10 or more cells that are used for the analysis should be described in detail. Furthermore, it’s helpful to show the original pictures used for the analysis either in the main figure or the supplementary. In this way, the data can be convincing to the audience.

Third, Figure 4 is the most interesting figure. However, there is only one representative to show the difference in invagination for wt and MLCP-BcLOV4. Some statistical data should be added to show that this phenotype is convincingly induced by MLCP-BcLOV4. For Fig 4a, it is also helpful to show the picture of the initial state (before illumination) for wt and MLCP-BcLOV4. In this way, a complete picture of the story can be shown to the audience. Another confusing point is in the legend of Figure 4c, it said “the fluorescence intensity of mCherry signal”, but there is no mCherry in the wt, right?

Above are the three major points concerned. I believe that addressing these three points could improve the paper quality and make it decent to be accepted for publication.

Below are minor points:

Line 84, engineer to engineered

Line 86, usually to are usually

Line 89, LOV domain is belonging to PAS domain superfamily, please double check to see if modification needed

Line 93, removed to was removed

Line 102, named to was named

In Figure 1d, how long is the 488 nm illumination performed? How is the light intensity? Normally the absolute light power is important information for optogenetic tools, but it might be difficult to measure this from the laser equipped to the microscope. However, do you have the information about the power of the laser at least? Also in the picture of Fig 1d, the labeled MYPT should be PP1C::MYPT to be complete, right?

Line 151, delete “of”

Line 174, contacting to contact / or delete “with”

Lie 200, epidermis specific to epidermis-specific

Line 279, accurate to accuracy

Line 347, gluteraldehyde to glutaraldehyde

Line 355, needed of to needed for

Line 363, exposure to exposed

Author Response

Reviewer 1: Comments and Suggestions for Authors

Optogenetic method utilized to manipulate subcellular protein activities is of great interest. The authors of this manuscript developed a new optogenetic system to manipulate the MLCP activity and thus suppress actomyosin activity in Ciona epidermis. The overall story is very interesting except there are several points which need to be improved.

First, as already mentioned by the authors, there is already a similar system using iLID to manipulate MLCP activity. A question to the audience might be, why the authors do not test the already established optogenetic system based on iLID in Ref 45 but develop a new system with BcLOV4 in this study. Obviously, there are several differences between iLID and BcLOV4, the differences should be introduced in more detail, this will help the audience to choose the potential tool for their own research.

Response: Thanks for your comments. Yes. The iLID system in Ref 45 has been used well in mammalian cells and Xenopusembryos. The reasons why we didn’t use this system in Ciona system are as following:

  1. The localization of iLID-GFP proteins could not be examined before the illumination, because of the wave length of GFP excitation light is similar with that of iLID system (488 nm) in this system.
  2. We tested iLID-GFP system in our hand, and found that it was very light-sensitive and easily to be illuminated under the confocal microscopy.

While, there is only one construct in BcLOV4 system, in which mCherry could be introduced to fuse. Thus, the localization of the construct before illumination could be examined by 587 nm. Based on these experiences, we decided to develop a newly Ciona-suitable system based on the BcLOV4 system. We have added these interpretations in the revision (line 100-101).

Second, Figure 2e is somehow not very convincing as it shows in its current form. It is not very common to label p<0.1 as “ * “, which usually means significant. For such a result, it is also not very convincing to me with n=10. Since the cells should vary a lot even in their natural condition, how choosing the 10 or more cells that are used for the analysis should be described in detail. Furthermore, it’s helpful to show the original pictures used for the analysis either in the main figure or the supplementary. In this way, the data can be convincing to the audience.

Response: Sorry for the mistake. We rechecked the calculation of the t-test and found that the p value should be 0.0332 based on the original data. We now corrected it as “p < 0.05” in the revision.

By the way, the p value for Fig.4c was also wrong. It should be 0.035. We now corrected it as “p < 0.05” in the revision.

The number n = 10 here indicates the selected cell number, not all the experimental cell number. We actually conducted several times of experiments using approximately 50-60 cells, but in order to minimize the effect of the intercellular force, we only selected 10 cells for the calculation. We have added the original curves of circularity over time of these ten cells, and the representative original picture used for the analysis in the supplemental data Figure A1 in the revision. The corresponding text also has been added in the revision (line 177-178).

Third, Figure 4 is the most interesting figure. However, there is only one representative to show the difference in invagination for wt and MLCP-BcLOV4. Some statistical data should be added to show that this phenotype is convincingly induced by MLCP-BcLOV4. For Fig 4a, it is also helpful to show the picture of the initial state (before illumination) for wt and MLCP-BcLOV4. In this way, a complete picture of the story can be shown to the audience. Another confusing point is in the legend of Figure 4c, it said “the fluorescence intensity of mCherry signal”, but there is no mCherry in the wt, right? 

Response: Thanks for the suggestion. We now quantified the fluorescence ratio, and length ratio of the apical to basal of vertical section of atrial siphon apical contraction. The data was now added as supplementary data Figure A2a and Figure A2b, respectively. The additional experimental images of MLCP-BcLOV4 were shoed as supplemental data Figure A2C.

We are sorry for the mistake on Figure 4c legend. Here, the signal is not from mCherry. It is from the fluorescence intensity of active myosin II antibody staining. This has been corrected in the revision (line 269).

Above are the three major points concerned. I believe that addressing these three points could improve the paper quality and make it decent to be accepted for publication.

Below are minor points:

Line 84, engineer to engineered

Response: Thank you for the correction.

Line 86, usually to are usually

Response: Thank you for the correction.

Line 89, LOV domain is belonging to PAS domain superfamily, please double check to see if modification needed

Response: Thanks for your comments, we have follow your suggestion and modified in the text.

Line 93, removed to was removed

Response: Thanks for your correction.

Line 102, named to was named

Response: Thanks for your correction.

In Figure 1d, how long is the 488 nm illumination performed? How is the light intensity? Normally the absolute light power is important information for optogenetic tools, but it might be difficult to measure this from the laser equipped to the microscope. However, do you have the information about the power of the laser at least? Also in the picture of Fig 1d, the labeled MYPT should be PP1C::MYPT to be complete, right?

Response: We performed the illumination for 3 mins. This information has been added in Figure 1d.

The nominal power of diode laser 488nm was 30mW. As you indicated we don’t know the light intensity in our experiments. 

For Fig. 1d, the labeling of MYPT here is correct, we linked PP1C with MYPT after this experiment.

Line 151, delete “of”

Line 174, contacting to contact / or delete “with”

Lie 200, epidermis specific to epidermis-specific

Line 279, accurate to accuracy

Line 347, gluteraldehyde to glutaraldehyde

Line 355, needed of to needed for

Line 363, exposure to exposed

Response: Thanks for all the above correction. We have made correction in the text.

Reviewer 2 Report

This paper is very well written and suits the IJMS scientific profile. As a mechanobiology researcher, I have read that paper with excitation. Here I have a couple of questions:
I'm not an ocean-related researcher. Why you used the Ciona system in your study? Is that a typical model system in marine genetics?
You said that MLCP-BcLOV4 system is a powerful tool for the study of biomechanical mechanisms. What specific applications do you mean? Could you elaborate on that in the Discussion section?

Author Response

Reviewer 2

This paper is very well written and suits the IJMS scientific profile. As a mechanobiology researcher, I have read that paper with excitation. Here I have a couple of questions:
I'm not an ocean-related researcher. Why you used the Ciona system in your study? Is that a typical model system in marine genetics?
You said that MLCP-BcLOV4 system is a powerful tool for the study of biomechanical mechanisms. What specific applications do you mean? Could you elaborate on that in the Discussion section?

Response: Thanks very much for your comments and interests on this work. Ciona, as the marine model animal has been used for hundred years for EvoDevo study. Ciona system is special for organ morphogenesis study because of its simple tissue structure and small cell number. The MLCP-BcLOV4 system we constructed can regulate the activity and the polarized distribution of myosin II at the subcellular level. Our work demonstrated that this system can interfere the polarity of the phosphorylated myosin II in apical contraction during Ciona atrial siphon morphogenesis and lead to failure of epidemical cell invagination. In Ciona, there are many similar morphogenetic processes, in which the polarized distribution of actin is involved. Such as the morphogenesis of Ciona embryonic notochord and the endodermal invagination process. We believe that our system can be used in these developmental processes to advance the study of organ morphogenesis.

Round 2

Reviewer 1 Report

The authors addressed all the points that I was concerned about. I recommend the paper be accepted for publication.